# The Use of Salvage Chemotherapy for Patients with Relapsed Testicular Germ Cell Tumor (GCT) in Canada: A National Survey

Esmail M. Al-Ezzi [1], Amer Zahralliyali [1], Aaron R. Hansen [1,2], Robert J. Hamilton [3], Michael Crump [1], John Kuruvilla [1], Lori Wood [4], Lucia Nappi [5], Christian K. Kollmannsberger [5], Scott A. North [6], Eric Winquist [7], Denis Soulières [8], Sebastien J. Hotte [9] and Di Maria Jiang [1,*]

[1] Division of Medical Oncology and Hematology, Princess Margaret Cancer Centre, University Health Network, University of Toronto, Toronto, ON M5S 1A1, Canada; esmail.al-ezzie@uhn.ca (E.M.A.-E.); dr.amerzahr84@gmail.com (A.Z.)

[2] Division of Cancer Services, Princess Alexandra Hospital, Metro South Health, Brisbane, QLD 4113, Australia

[3] Division of Urology, Department of Surgery, Princess Margaret Cancer Centre, University Health Network, University of Toronto, Toronto, ON M5S 1A1, Canada

[4] Queen Elizabeth II Health Sciences Centre, Dalhousie University, Halifax, NS B3H 4R2, Canada

[5] Department of Medicine, British Columbia Cancer Agency, University of British Columbia, Vancouver, BC V6T 1Z1, Canada

[6] Division of Medical Oncology, Cross Cancer Institute, University of Alberta, Edmonton, AB T6G 2R3, Canada

[7] Department of Oncology, London Health Sciences Centre, Western University, London, ON N6A 3K7, Canada

[8] Département Hématologie-Oncologie, Centre Hospitalier de l'Université de Montréal, Montréal, QC H2X 0C1, Canada

[9] Juravinski Cancer Centre, McMaster University, Hamilton, ON L8S 4L8, Canada

* Correspondence: di.jiang@uhn.ca; Tel.: +1-416-946-4501 (ext. 4807)

**Abstract:** Background: Although metastatic germ cell tumor (GCT) is highly curable with initial cisplatin-based chemotherapy (CT), 20–30% of patients relapse. Salvage CT options include conventional (CDCT) and high dose chemotherapy (HDCT), however definitive comparative data remain lacking. We aimed to characterize the contemporary practice patterns of salvage CT across Canada. Methods: We conducted a 30-question online survey for Canadian medical and hematological oncologists with experience in treating GCT, assessing treatment availability, patient selection, and management strategies used for relapsed GCT patients. Results: There were 30 respondents from 18 cancer centers across eight provinces. The most common CDCT regimens used were TIP (64%) and VIP (25%). HDCT was available in 13 centers (70%). The HDCT regimen used included carboplatin and etoposide for two cycles (76% in 7 centers), three cycles (6% in 2 centers), and the TICE protocol (11%, in 2 centers). "Bridging" CDCT was used by 65% of respondents. Post-HDCT treatments considered include surgical resection for residual disease (87.5%), maintenance etoposide (6.3%), and surveillance only (6.3%). Conclusions: HDCT is the most commonly used GCT salvage strategy in Canada. Significant differences exist in the treatment availability, selection, and delivery of HDCT, highlighting the need for standardization of care for patients with relapsed testicular GCT.

**Keywords:** relapsed germ cell tumors; salvage chemotherapy; conventional-dose chemotherapy; high-dose chemotherapy; autologous stem cell transplant

## 1. Introduction

Germ cell tumors (GCT) are the most common cancer affecting young males between 20 and 34 years of age. An estimated 1200 new cases and 35 deaths occurred in Canada in 2021 due to GCT [1].

Although metastatic GCT is highly curable, 20–30% of patients relapse after initial cisplatin-based chemotherapy, including 40–50% of patients presenting with International

Germ Cell Cancer Collaborative Group (IGCCG) poor-risk disease [2]. For these relapsed patients, salvage chemotherapy is associated with long-term cure rates of 50% [3]. Salvage chemotherapy options include conventional-dose (CDCT) and high-dose chemotherapy (HDCT) with autologous stem cell transplant (ASCT); however, treatment strategies vary in the absence of definitive comparative data, pending results from TIGER clinical trial. The TIGER trial is a randomized phase III trial of CDCT (paclitaxel, ifosfamide and cisplatin (TIP)) vs. HDCT (two cycles of paclitaxel and ifosfamide followed by three cycles of high-dose carboplatin and etoposide (TI-CE)) as initial salvage chemotherapy for patients with GCT (NCT02375204) [4]. Currently, IT-94 is the only randomized clinical trial which has compared four cycles of CDCT (cisplatin, etoposide, and ifosfamide (VIP) or vinblastine, ifosfamide, and cisplatin (VeIP) (arm A) to three cycles of CDCT (VIP or VeIP), followed by one cycle of HDCT (carboplatin, etoposide, and cyclophosphamide) with ASCT (arm B), with no significant difference in overall survival (OS) [5]. Several methodological shortcomings make the interpretation of this trial's results challenging, with the exclusion of patients who did not achieve complete or partial response to first line chemotherapy, a significant proportion of patients (>25%) not receiving planned HDCT, and only one HDCT cycle being used.

For relapsed GCT patients requiring salvage chemotherapy, the International Prognostic Factor Study Group (IPFSG) risk stratification is prognostic and may potentially be a useful tool for treatment selection. The IPFSG criteria include the primary site of disease, prior initial treatment response, the progression-free interval, AFP and B-HCG levels at relapse, and the presence of distant metastasis such as liver, brain, and bone. Results from a large, multicentered, retrospective database reported by Lorch et al. suggest the superiority of HDCT over CDCT in progression-free survival (PFS) and OS in each IPFSG subgroup except for low-risk patients. However, inherent limitations exist due to the retrospective nature of this study [5–7]. Other retrospective data suggest IPFSG very-low-risk and low-risk patients have similar outcomes with CDCT or HDCT [8], although prospective studies are warranted for validation.

Given the uncertainty of the optimal curative-intent salvage chemotherapy strategy in this setting and the potential for improving outcomes, we aimed to characterize the contemporary real-world practice patterns of salvage chemotherapy across Canada, including treatment availability, patient selection, and management strategies used for relapsed GCT patients.

## 2. Materials and Methods

This was a multicenter national survey of physicians to examine real-world practice patterns of the use of salvage chemotherapy in patients with relapsed GCT. To our knowledge, no prior surveys on this topic were published in the literature at the time of study design. The survey questions were developed with multidisciplinary input and revisions from medical oncologist and hematologists who treat relapsed GCT. The survey was pilot tested to ensure the questions were clearly articulated, relevant, and comprehensive. The final online survey consisted of 30 multiple choice questions (Supplementary File S1) and took approximately 7 min to complete. The survey was administered via Survey Monkey in August 2021, using email membership listservs of Genitourinary Medical Oncologists of Canada and Cell Therapy Transplant Canada to capture both staff medical oncologists and hematologists. Participation was voluntary with no financial incentive to complete the survey. Reminders emails were sent every 2–4 weeks after the initial email invitations, and the survey was closed in March 2022. Respondents could choose not to respond to certain questions. The survey was anonymous, no survey respondent was identified individually, and all data were reported in aggregate and confidentially. Email addresses were discarded at the conclusion of this study. Data were analyzed using quantitative methodologies and descriptive statistics. Response rates and other categorical data were reported as proportions, while continuous data were described using range. Free text responses were categorized to reveal any notable trends. All percentages were calculated as a function of

the number of respondents for each question. Data from incomplete questionnaires were included for analysis whenever possible.

## 3. Results

The respondents included 30 physicians, 25 (83.3%) medical oncologists, 3 (10%) hematologists, 2 (6.6 %) identified as both. Responses were captured from 18 cancer centers across eight provinces: Ontario (53.3%); Quebec (20%); Alberta, (6.6%); British Columbia, (3.3%); Manitoba, (6.6%); New Brunswick, (3.3%); Nova Scotia, (3.3%); and PEI, (3.3%) (Table 1).

**Table 1.** Characteristics of Survey Respondents and Programs.

| Question | Answers | *n* = 30 | Total Response | % |
|---|---|---|---|---|
| Specialty | Medical Oncology | 25 | 30 | 83.3 |
| | Malignant Hematology | 3 | | 10 |
| | Both | 2 | | 6.7 |
| | Staff | 30 | | 100 |
| Cancer Center location | Ontario | 16 | 30 | 53.3 |
| | Quebec | 6 | | 20 |
| | Alberta | 2 | | 6.6 |
| | BC | 1 | | 3.3 |
| | Manitoba | 2 | | 6.6 |
| | New Brunswick | 1 | | 3.3 |
| | Nova Scotia | 1 | | 3.3 |
| | PEI | 1 | | 3.3 |
| Number of patients receiving HDCT + ASCT for relapsed GCT at your center | <1 cases/year | 3 | 20 | 15 |
| | 1 case/year | 3 | | 15 |
| | 1–5 cases/year | 11 | | 55 |
| | 6–10 cases/year | 3 | | 15 |

Abbreviations: BC, British Columbia; PEI, Prince Edward Island; ASCT, autologous stem cell transplant; GCT, germ cell tumor.

Most respondents (26, 86%) were from academic centers, and reported case volumes of five salvage chemotherapy cases or less per year (Table 1). No clinical trials were available for relapsed GCT patients requiring salvage chemotherapy during the time of this survey. The most commonly used CDCT regimen was TIP (66.6%) and VIP (23.3%) (Figure 1).

HDCT was available for 72.4% of respondents in 13 centers. HDCT regimen used included carboplatin and etoposide for 2 cycles with tandem ASCT (Indiana protocol, 77.8%) and the TICE protocol 11.1%) (Figure 2). Planned hospital admission was required by 83.3% of the respondents who offered salvage HDCT and ASCT.

Among 20 respondents who offered HDCT and ASCT, only 25% reported HDCT having been organized within three weeks. Most (65.0%) required CDCT as a bridging therapy while waiting for HDCT and ASCT to be organized. Stem cells were most often collected within four weeks (69.2%) with a minimum target CD34 cell dose of $2\text{–}3 \times 10^6$/kg (33.3%), and salvage HDCT and ASCT would begin 2–4 weeks after peripheral stem cell collection (66.6%). After the completion of bridging CDCT, most (61.5%) required both tumor markers (AFP, BHCG, LDH) and radiological imaging to assess the response; 69.2% of those would have patients proceed to HDCT and ASCT regardless of the biochemical and radiology response. However, 7.7% would proceed only if there was evidence of disease response, and 23.1% would make the decision to proceed as per case-by-case discussion (e.g., one described giving another cycle of CDCT if there was no response). Others

mentioned discussion via a national email tumor board). After one cycle of HDCT, both tumor markers and radiology imaging were required to document response in 18.7% of the respondents, while 75% only required tumor markers. However, 37.5% of the respondents would proceed to the second transplant regardless of the results.

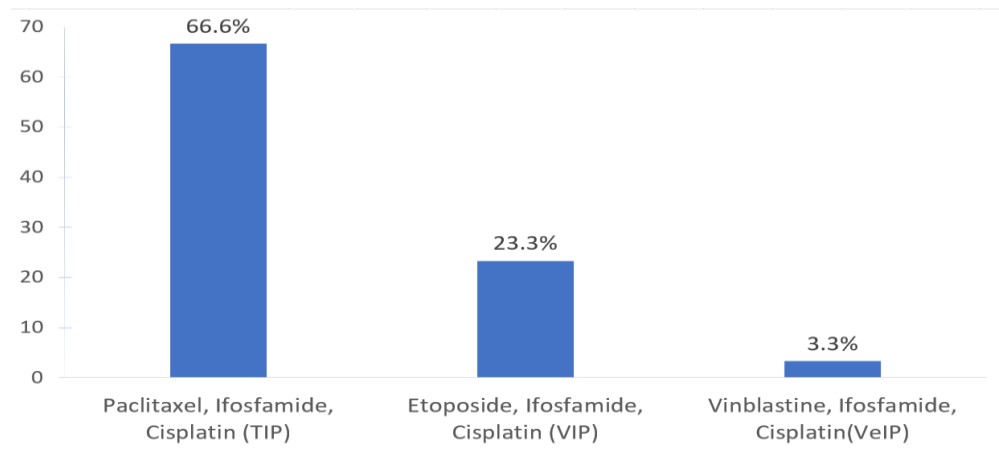

**Figure 1.** Most commonly used salvage conventional dose chemotherapy (CDCT) regimens.

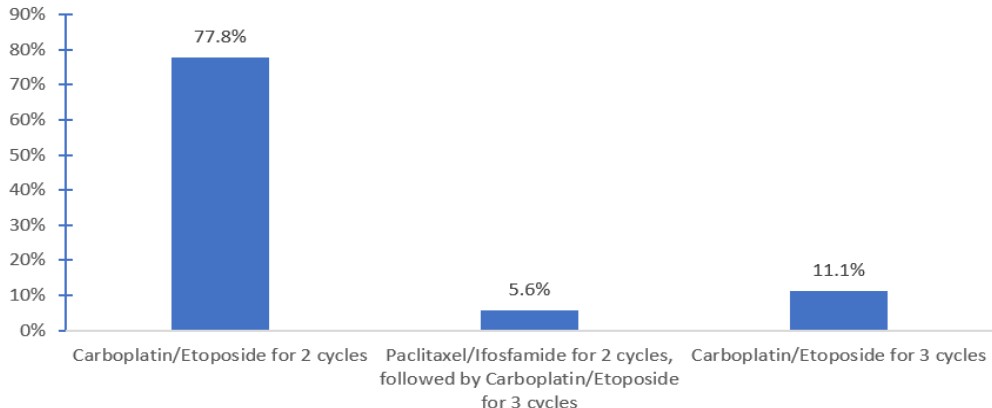

**Figure 2.** Most commonly used salvage high dose chemotherapy (HDCT) regimens.

With respect to treatment selection, most offered salvage HDCT and ASCT in the first-line salvage setting (75.9%); some favored second (20.7%) or third line (3.5%) settings. Only about a third of oncologists used the IPFSG criteria to determine eligibility for HDCT. IPFSG criteria did not impact treatment selection for HDCT for 37.9% (use HDCT regardless of IPFSG category); 24% of the respondents use CDCT only for very-low- and low-risk IPFSG disease; and 14% use CDCT for only low-risk IPFSG. Among the respondents, 17.2% were not familiar of the IPFSG criteria (Figure 3).

None reported any impact of the ongoing COVID-19 pandemic on selecting salvage chemotherapy (either CDCT or HDCT) for patients with relapsed GCT.

Surveillance investigations after the completion of HDCT and ASCT include tumor markers and imaging every 2–4 months in the first year (Table 2).

Patients are generally followed by medical oncology (56.2%) or both medical oncology and hematology (37.5%). Post-salvage chemotherapy treatment options offered after completion of HDCT and ASCT include surgical resection of the residual tumor if feasible (87.5%, 9 centers), maintenance etoposide (6.3%, one center), and none other than surveillance (6.3%, one center) (Figure 4).

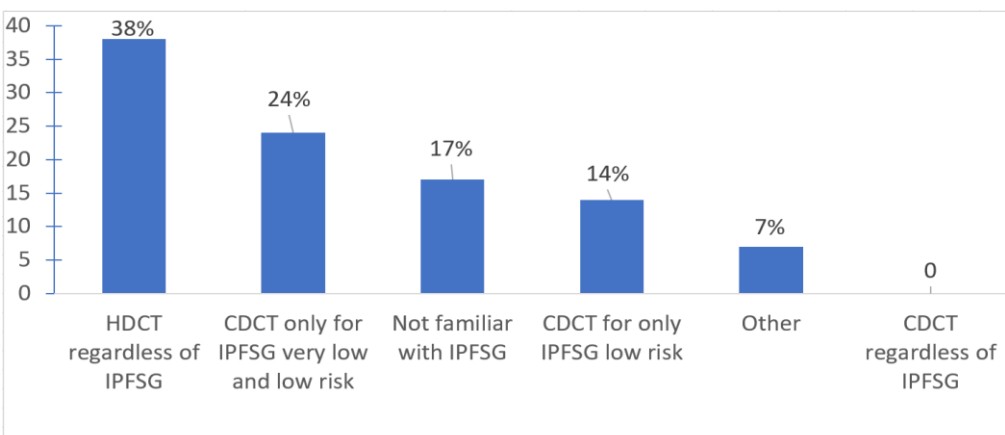

**Figure 3.** Treatment selection based on IPFSG risk classification according to the survey respondents. Abbreviations: IPFSG, International Prognostic Factors Study Group; HDCT, high-dose chemotherapy; CDCT, conventional-dose chemotherapy. Other: they discuss all cases provincially.

**Table 2.** Summary of the key questions and answers.

| Questions | Responses | *n* | Total Response | % |
|---|---|---|---|---|
| 1. Which of the following salvage chemotherapy treatments do you oversee at your center? | Use of CDCT only | 14 | 30 | 46.6 |
| | Use both CDCT and HDCT | 12 | | 40 |
| | Use of HDCT & ASCT only | 4 | | 13.3 |
| 2. Number of patients receiving salvage chemotherapy (CDCT, or HDCT) for relapsed GCT at your center | <1 cases/year | 5 | 30 | 16.6 |
| | 1 case/year | 7 | | 23.3 |
| | 1–5 cases/year | 12 | | 40 |
| | 6–10 cases/year | 6 | | 20 |
| 3. Percentage of salvage HDCT + ASCT given in the following treatment settings (% reflect averages of responses for each) (range) | First-line setting | 69% (0–100) | | |
| | Second-line salvage setting | 33% (0–100) | | |
| | Third-line salvage setting or beyond | 4% (0–20) | | |
| 4. If salvage HDCT + ASCT is not available at your center, when do you typically refer patients with relapsed germ cell tumors for salvage HDCT + ASCT? | Upon the first relapse after first line of cisplatin | 6 | 8 | 75 |
| | Upon further relapse after salvage CDCT | 1 | | 12.5 |
| | I do not usually refer a patient | 1 | | 12.5 |
| 5. Is "bridging" CDCT given while waiting for HDCT + ASCT? | No. HDCT can be organized in 3 weeks | 5 | 20 | 25 |
| | No. HDCT takes 3–6 weeks to organize, but no "bridging" CDCT is used | 2 | | 10 |
| | Yes | 13 | | 65 |
| 6. Is disease response (biochemical and/or radiographic) to" bridging" CDCT required to proceed with salvage HDCT + ASCT at your center? | Always. Patients receive HDCT + ASCT only if evidence of disease response. | 1 | 13 | 7.7 |
| | Never. Patients proceed to HDCT + ASCT regardless | 9 | | 69.2 |
| | Case by case discussion | 3 | | 23.1 |
| 7. When do you initiate apheresis/collection after completion of "bridging" CDCT? | Within 4 weeks | 9 | 13 | 69.2 |
| | Within 4–6 weeks | 1 | | 6.9 |
| | Within 6–8 weeks | 1 | | 6.9 |
| | I do not know | 2 | | 15.3 |

**Table 2.** *Cont.*

| Questions | Responses | *n* | Total Response | % |
|---|---|---|---|---|
| 8. Minimum number of collected CD34 cells required for salvage HDCT + ASCT to proceed | CD34+ cell count 2–3 $\times 10^6$/kg | 6 | 18 | 33.3 |
| | CD34+ cell count 3.1–4 $\times 10^6$/kg | 1 | | 5.6 |
| | CD34+ cell count 4.1–6 $\times 10^6$/kg | 0 | | 0 |
| | CD34+ cell count >6 $\times 10^6$/kg | 1 | | 5.6 |
| | I do not know | 10 | | 55.5 |
| 9. Initiation ASCT after peripheral stem cells collection | Within 2 weeks | 6 | 18 | 33.3 |
| | Within 2–4 weeks | 6 | | 33.3 |
| | Within 4–6 weeks | 1 | | 5.5 |
| | I do not know | 5 | | 27.7 |
| 10. Salvage HDCT + ASCT required planned admission to hospital | Yes | 15 | 18 | 83.3 |
| | No | 3 | | 16.7 |
| 11. Do the tumor markers and CT results post-first-cycle of HDCT + ASCT affect your decision to proceed with subsequent cycle of HDCT? | Yes. If disease progression, subsequent cycle of HDCT is abandoned. | 6 | 16 | 37.5 |
| | No. Patient proceeds with subsequent cycle of HDCT regardless. | 6 | | 37.5 |
| | Case-by-case | 4 | | 25 |
| 12. Surveillance investigations after completion of salvage HDCT + ASCT within the first year. | Tumor markers every 3 months | 16 | 24 | 50 |
| | Imaging every 4 months | 16 | | 50 |

Abbreviations: CDCT, conventional-dose chemotherapy; HDCT, high-dose chemotherapy; ASCT, autologous stem cell transplant.

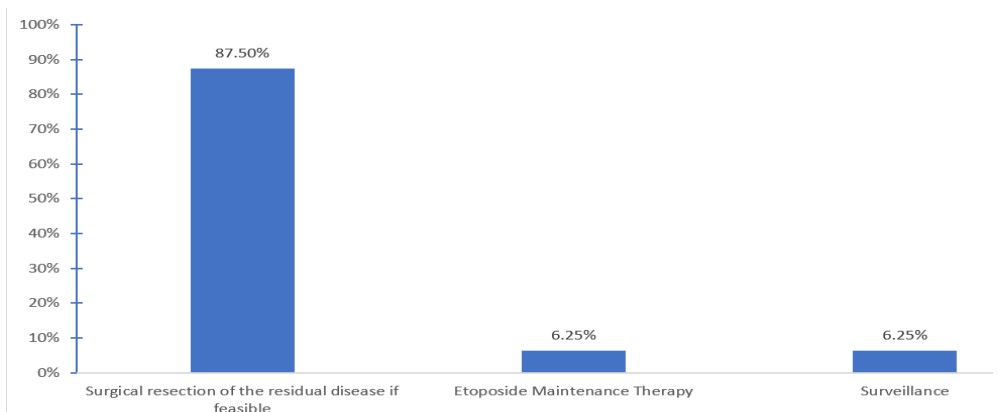

**Figure 4.** Treatment options offered after completion of the salvage HDCT + ASCT.

## 4. Discussion

For patients with relapsed GCT following initial cisplatin-based chemotherapy, the optimal salvage chemotherapy regimen is still being debated. According to this Canadian survey study, the use of salvage HDCT and ASCT was favored over CDCT in the first line salvage setting upon relapse following initial cisplatin-based chemotherapy, which is consistent with previous data [9]. The IPFSG criteria was used by about a third of oncologists surveyed to select CDCT for IPFSG very low risk and/or low risk patients, and 17% surveyed were in fact not familiar with this prognostic score. Our results echo the observed variation of treatment patterns around the world. Investigators from Indiana University favor the use of HDCT as the initial salvage therapy approach [10]. In Germany, Zschäbitz et al. published their experience in selecting HDCT as the initial salvage therapy approach in 67% of patients, including IPFSG intermediate- (*n* = 12, 26%), high- (*n* = 16, 35%), or very-high-risk (*n* = 9, 20%) relapsed GCT [11]. Oncologists at Memorial Sloan Kettering employ a risk-stratified approach of CDCT with TIP or HDCT with the TICE

protocol [12]. In comparison, in the UK, salvage HDCT is mostly reserved for the third-line setting [13].

Large, retrospective, multicentered data have suggested the potential utility of IPFSG criteria in treatment selection [8], which has been validated in multiple other cohorts. Available data suggest that HDCT is associated with a 10–15% improvement in OS rates compared to CDCT in all risk groups except low-risk IPFSG disease, which awaits confirmation from the TIGER trial. This trial aims to compare overall survival with TIP versus a TI–CE regimen (1:1), stratified by IPFSG risk classification (low, intermediate, and high) (clinicaltrials.gov identifier NCT02375204). Based on our survey results, in Canada, both approaches of using CDCT in IPFSG very-low- or low-risk groups or HDCT in all IPFSG risk groups are used and are acceptable options. The use of HDCT in a first-line salvage setting may be preferable to some, given that it is more effective than its use in subsequent treatment settings [14].

Notably, the 2010 Canadian consensus guidelines for the management of testicular GCT indicated that either CDCT or HDCT were options in patients who relapse post-cisplatin-based chemotherapy and that a number of clinical and tumor-related factors must be taken into account when choosing [15]. The updated 2022 Canadian consensus guidelines do recommend risk stratifying patients based on the IPFSG risk stratification but caution that there is no prospective conclusive randomized data identifying which type of salvage treatment produces the best outcomes. It recommends that in patients with very low or low IPFSG risk, either CDCT or HDCT are reasonable options; in patients in the intermediate-IPFSG-risk group, HDCT is the preferred treatment option; however, there may be some circumstances in which CDCT is reasonable, and for patients with high and very high IPFSG risk, HDCT is the preferred treatment option [16].

Most commonly, salvage HDCT was administered as a tandem high-dose carboplatin etoposide with ASCT (Indiana protocol), which is associated with 2-year PFS and 5-year OS rates of 47–65% [6,14,17,18]; however, two (11%) used the TICE regimen. In most centers, bridging CDCT was required due to the logistics of organizing HDCT. Whether this improves the effectiveness of salvage HDCT is unknown and requires further study. It is notable that some physicians would offer salvage HDCT only if there were a disease response to CDCT. In the few cases where patients do not respond to bridging CDCT, some can still achieve curing, and current data suggest they should be excluded from HDCT [19]. On the other hand, some respondents indicated that patients will still proceed with the subsequent cycle of HDCT if the disease progresses after the first cycle. To our knowledge, there is no evidence supporting this approach and it is unlikely that long-term disease control will be achieved in this situation, considering the potential toxicity, cost, and limited benefit. Instead, alternative strategies such as post-chemo surgical resection for nonseminoma, or alternative palliative chemotherapy regimens such as GemOx should be considered [20].

The most common conventional CDCT regimen used in our survey is TIP followed by VIP. Many phase II studies support using TIP as the preferred initial salvage CDCT regimen. Complete remission rates with TIP have been reported to be between 31–71%. The MSKCC group have shown 2-year PFS rates of 65% and 2-year OS rates of 78%, which represent some of the best outcomes of salvage chemotherapy with CDCT. It is important to note that this study reported on a select group of patients who had somewhat favorable disease characteristics including purely gonadal primaries and response to initial cisplatin-based chemotherapy for ≥6 months prior to relapse [21–23]. Other retrospective and prospective phase II studies evaluating VIP or VeIP have shown CR rates in the range of 19–56%, which supports their use as less favored but valid alternative options of CDCT [24–29]. After the completion of bridging CDCT, only (61.5%) of the respondents required both tumor markers (AFP, BHCG, LDH) and radiological images. Similarly, investigations including tumor markers and imaging studies after the first cycle of HDCT can lead to very different treatment decisions. The suggests that the performance and interpretation of these tests are not uniform across centers, and that there is a need for more data to guide clinical decision-

making, as well as a need for the harmonization of testing protocols to further optimize care in this setting. At our center, we measure tumor markers prior to each cycle of salvage chemotherapy to evaluate major trends, using validated marker assays including detection of bHCG. We interpret tumor marker measurements along with radiographic evaluation, and thus do not make treatment decisions based on low levels or minor fluctuations of markers alone during salvage chemotherapy [30,31].

In our survey, most considered surgical resection of residual disease feasible post-salvage chemotherapy (87.5%, nine centers). This is supported by the most recent Canadian guideline recommendations [16]. Cary et al. reported the outcomes of the post- HDCT and ASCT retroperitoneal lymph node dissection (RPLND) in 92 patients treated at Indiana University. Histological findings were viable tumor (38%), necrosis (26%), and teratoma (34%). The 5-year OS of patients undergoing RPLND after HDCT was 70% in the Indiana University cohort [32]. Miller et al. reported the surgical outcomes of 112 patients who underwent RPLND following salvage CDCT or HDCT at Memorial Sloan Kettering Cancer Center. Histopathological findings were viable cancer (27%), teratoma (23%), and fibrosis (50%) [12]. Maintenance oral etoposide (50 mg po daily 21 days every 28 days) post salvage therapy is not routinely used in Canada. Retrospective data have shown potential promise for this approach [33,34]. However, further prospective data are needed. A randomized phase II clinical trial of maintenance oral etoposide or observation following HDCT for relapsed GCT is ongoing and pending results (NCT04804007) [35].

We acknowledge several limitations of our study. Patients with relapsed GCT who require salvage chemotherapy are uncommon, and the low case volumes of some centers may have skewed some of the results. Centralization of care for these patients can be challenging considering the vast geography of Canada, and future efforts in standardization of practice across regions will be critical. The low response rate of hematologists limits the data on the current delivery of HDCT and ASCT in Canada; however, building on this survey study, differences in institutional protocols and resources for transplant in this rare treatment setting will be investigated in future studies.

## 5. Conclusions

HDCT was available in most academic centers in Canada and was the most commonly preferred strategy in a first-line salvage treatment setting. Two cycles of carboplatin and etoposide is the most common HDCT regimen. TIP is the most commonly used CDCT regimen. Significant differences exist in the treatment selection, delivery of HDCT, and post-salvage chemotherapy care, highlighting the need for the standardization of care for patients with relapsed testicular GCT requiring salvage chemotherapy. Patients who relapse after initial chemotherapy require multidisciplinary expertise at experienced centers for optimal and timely management.

**Supplementary Materials:** The following are available online at https://www.mdpi.com/article/10.3390/curroncol30070458/s1, File S1: The Use of Salvage Chemotherapy for Patients with Relapsed Testicular Germ Cell Tumour in Canada: A National Survey Study Questions.

**Author Contributions:** Conceptualization, E.M.A.-E., A.Z., A.R.H. and D.M.J.; Data curation, E.M.A.-E.; Formal analysis, E.M.A.-E., A.Z. and D.M.J.; Methodology, E.M.A.-E., A.Z. and D.M.J.; Resources, E.M.A.-E., A.R.H., R.J.H., M.C., J.K., L.W., L.N., C.K.K., S.A.N., E.W., D.S., S.J.H. and D.M.J.; Software, E.M.A.-E., A.Z. and D.M.J.; Supervision, D.M.J.; Writing—original draft, E.M.A.-E., A.Z., R.J.H., M.C., L.W. and D.M.J.; Writing—review and editing, E.M.A.-E., A.R.H., R.J.H., M.C., J.K., L.W., L.N., C.K.K., S.A.N., E.W., D.S., S.J.H. and D.M.J. All authors have read and agreed to the published version of the manuscript.

**Funding:** This research received no external funding.

**Institutional Review Board Statement:** This study was a survey study of clinicians which did not require research ethics board review.

**Informed Consent Statement:** Not applicable.

**Data Availability Statement:** The data that support the findings of this study are available on request from the corresponding author. The data are not publicly available due to privacy or ethical restrictions.

**Acknowledgments:** We would like to express our special thanks to our colleges Carlos Stecca, Philippe Bedard and Srikala Sridhar for their insightful advice and suggestions which were especially valuable to us for this project. We thank the Meekison, Keystone and Posen families for their support for the Canadian GCT symposium.

**Conflicts of Interest:** R.J.H. received fund from Janssen, Abbvie, Bayer, Astellas Pharma (Honoraria); Janssen, Bayer (Research Funding); Janssen (Travel, Accommodations, Expenses). A.R.H. received funding from Merck, GlaxoSmithKline, Bristol-Myers Squibb, Eisai, Novartis AstraZeneca (Consulting or Advisory Role); and Karyopharm Therapeutics, Merck, Bristol-Myers Squibb, Boehringer Ingelheim, GlaxoSmithKline, Roche/Genentech, Janssen, AstraZeneca/MedImmune, Astellas Pharma, BioNTech, Pfizer/EMD Serono, Neoleukin Therapeutics (Research Funding). M.C. received fund from Gilead Sciences, Servier/Pfizer (Honoraria); SERVIER, Gilead Sciences, Novartis Canada Pharmaceuticals Inc (Consulting or Advisory Role); Roche Canada (Research Funding). L.W. received fund from Bristol-Myers Squibb, Pfizer, Roche Canada, Merck, AstraZeneca (Research Funding). C.K.K. received fund from Pfizer, Bristol-Myers Squibb, Ipsen, Merck KGaA, Merck, Astellas Pharma, Janssen Oncology, Eisai (Honoraria); Pfizer, Bristol-Myers Squibb, Astellas Pharma, Ipsen, Eisai, Janssen, Merck KGaA, Merck, Gilead Sciences (Consulting or Advisory Role); Pfizer, Ipsen (Travel, Accommodations, Expenses). S.A.N. received fund from Janssen-Ortho, Astellas Pharma, Pfizer, Sanofi, Merck, Roche Canada, Bristol-Myers Squibb (Honoraria); Astellas Pharma, Sanofi, Pfizer, Roche Canada, Merck, AstraZeneca, Janssen, Bristol-Myers Squibb, EMD Serono (Consulting or Advisory Role); Astellas Pharma, Janssen, AstraZeneca (Research Funding); AstraZeneca, Astellas Pharma (Travel, Accommodations, Expenses). E.W. received fund from Merck, Bayer, Eisai, Amgen, Roche, Ipsen (Consulting or Advisory Role), Roche/Genentech, Merck, Pfizer, Eisai, Ayala Pharmaceuticals (Research Funding). D.S. received fund from Merck, Novartis, Pfizer, AstraZeneca, Ipsen, Bristol-Myers Squibb, Eisai, Adlai Nortye (Honoraria); Merck, Pfizer, Ipsen, Adlai Nortye(Consulting or Advisory Role); Novartis, Pfizer, Merck, Roche/Genentech, Bristol-Myers Squibb, Lilly, Adlai Nortye(Research Funding). S.J.H. received fund Astellas Scientific and Medical Affairs Inc, Janssen Oncology, Bayer, Merck Ipsen, Eisai (Honoraria); Janssen, Astellas Pharma, Bristol-Myers Squibb, Pfizer, Bayer AstraZeneca, Merck, Eisai, Ipsen, Advanced Accelerator Applications Seattle Genetics (Consulting or Advisory Role); Ayala Pharmaceuticals, Pfizer, Astellas Pharma, Bristol-Myers Squibb-Seattle Genetics, SignalChem (Research Funding); Eisai, (Travel, Accommodations, Expenses). D.M.J. received fund from Janssen Oncology, Ipsen, Bayer, EMD Serono (Honoraria); Bayer (Consulting or Advisory Role). All other authors have no COI.

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
