# Peer review of "The Use of Salvage Chemotherapy for Patients with Relapsed Testicular Germ Cell Tumor (GCT) in Canada: A National Survey"

_curroncol, doi:10.3390/curroncol30070458_

Round 1
Reviewer 1 Report
This paper is relevant to highlight the poor standardization of management for patients with relapsed testicular GCT.
Major comments:
1. The authors should consider also the poor harmonization of protocols.
2. In this paper the authors have found that"After completion of bridging CDCT, most (61.5%) required both tumor markers (AFP, BHCG, LDH)". I think that the poor use of biomarkers to monitor chemotherapeutic success should be critically appraised in the discussion. Several papers have been focused on two main issues concerning the measurement of HCG.
The first issue concerns the type of assay which should be used for application in oncology. Refer to the following paper to highlight this issue. Ferraro S, Trevisiol C, Gion M, Panteghini M. Human Chorionic Gonadotropin Assays for Testicular Tumors: Closing the Gap between Clinical and Laboratory Practice. Clin Chem 2018;64:270-278.
The second issue concerns the interpretation of HCG accounting for the possible hypogonadism due to chemotherapeutic treatment. Refer to the following paper to highlight this issue. Ferraro S, Incarbone GP, Rossi RS, Dolci A, Panteghini M. Human chorionic gonadotropin in oncology: a matter of tight (bio)marking. Clin Chem Lab Med 2020;58:e57-e60.
3 Spell out the acronyms for the first time(see the Abstract).
Reviewer 2 Report
I congratulate the Authors for they excellent paper.
I just have a couple of comments
In the Introduction chapter Authors mention the study IT-94, but please note that among inclusion criteria also patients with partial remission have been included.
The Authors do not mention the problem of late relapses which still nowadays is a very intriguing aspect. Maybe the question was not among the proposed ones. (please just comment on this)
Round 2
Reviewer 1 Report
No further comments
Reviewer 2 Report
Thank you to the Authors for providing their modifications.